# Comorbidity Patterns in Patients with Atopic Dermatitis Using Network Analysis in the EpiChron Study

**DOI:** 10.3390/jcm11216413

**Published:** 2022-10-29

**Authors:** Manuel Almenara-Blasco, Jonás Carmona-Pírez, Tamara Gracia-Cazaña, Beatriz Poblador-Plou, Juan Blas Pérez-Gilaberte, Alba Navarro-Bielsa, Antonio Gimeno-Miguel, Alexandra Prados-Torres, Yolanda Gilaberte

**Affiliations:** 1Department of Dermatology, Miguel Servet University Hospital IIS Aragon, Paseo Isabel la Católica 1-3, 50009 Zaragoza, Spain; 2EpiChron Research Group, Aragon Health Sciences Institute (IACS), IIS Aragón, Miguel Servet University Hospital, 50009 Zaragoza, Spain; 3Health Services Research on Chronic Patients Network (REDISSEC), Network for Research on Chronicity, Primary Care, and Health Promotion (RICAPPS), ISCIII, 28029 Madrid, Spain; 4Subdirección Técnica Asesora de Gestión de la Información, Andalusian Health Service, 41071 Seville, Spain; 5Department of Internal Medicine, Miguel Servet University Hospital IIS Aragon, 50009 Zaragoza, Spain

**Keywords:** atopic dermatitis, comorbidities, patterns, network analysis

## Abstract

Background: Atopic dermatitis (AD) is associated with different comorbidities. Methods: Retrospective, observational study based on clinical information from the individuals of the EpiChron Cohort Study (Aragon, Spain) with a diagnosis of AD between 1 January 2010 and 31 December 2018. We calculated the tetrachoric correlations of each pair of comorbidities to analyze the weight of the association between them. We used a cut-off point for statistical significance of *p*-value < 0.01. Results: The prevalence of AD in the EpiChron Cohort was 3.83%. The most frequently found comorbidities were respiratory, cardio-metabolic, cardiovascular, and mental health disorders. Comorbidities were combined into 17 disease patterns (15 in men and 11 in women), with some sex and age specificities. An infectious respiratory pattern was the most consistently described pattern across all ages and sexes, followed by a cardiometabolic pattern that appeared in patients over 18 years of age. Conclusions: Our study revealed the presence of different clinically meaningful comorbidity patterns in patients with AD. Our results can help to identify which comorbidities deserve special attention in these types of patients and to better understand the physio-pathological mechanisms underlying the disease associations identified. Further studies are encouraged to validate the results obtained in different clinical settings and populations.

## 1. Introduction

Atopic dermatitis (AD) is a chronic inflammatory skin disease of multifactorial etiology characterized by dry skin, itching, erythema, inflammation and eczema formation [1]. It is estimated to affect 15–30% of children and 2–10% of adults [2]. Its prevalence has increased worldwide in recent years due to lifestyle and environmental changes, varying widely according to age and geographical area and being more prevalent in developed countries [2,3].

Advances in the understanding of the etiopathogenesis of AD suggest that its genesis is due to the interaction of several factors that act together to produce the onset and chronification of the disease. An alteration of the barrier function of the skin stands out, behind which there are underlying immune mechanisms, as well as genetic and environmental factors. Structural and functional abnormalities of the epidermis, together with skin inflammation due to an altered immune response, are the cornerstones of the pathogenesis of AD [4,5].

AD is considered as the cutaneous manifestation of a systemic disorder that also gives rise to other pathologies, such as asthma, allergic rhinoconjunctivitis, etc. Some patients with AD have elevated blood levels of IgE and eosinophils. These levels are related to the severity of AD, and for this reason lower values are shown on debut [4].

The immunological mechanisms involved are being investigated; Those proposed to date include antigen-presenting cutaneous dendritic cells in the pathogenesis, and also the loss of immunosuppressive capacity of CD_4_^+^, CD_25_^+^ Treg cells [4].

As occurs in other diseases of the atopy spectrum, the predominance of Th2 cells over Th1 generates an immunological imbalance that aggravates the pathogenesis of AD, increases IgE and activates interleukins.

On the other hand, the importance of the integrity of the skin barrier should be highlighted; in recent decades, its dysfunction has been determined to be essential in the pathogenesis of AD. The structure of the skin barrier is complex. The most superficial layer of the epidermis is the stratum corneum, it is made up of proteins (filaggrin, loricin, involucrin); and by a layer of lipids composed of long-chain ceramides as the main component. The stratum corneum protects against environmental stimuli such as allergens, irritants, chemical and physical changes and infections, it also prevents trans-epidermal dehydration [5].

Family genetic studies have shown that AD is a hereditary disease. At the moment, the evidence points to chromosome 1q21 where the locus of the epidermal differentiation complex is located [4,5]. Of all the components of the skin barrier, filaggrin and its mutations are the ones that have shown the greatest association with AD. Filaggrin is a protein that interacts with intermediate filaments, producing their aggregation into macrofibrils. Filaggrin defects can lead to dysfunctions in the skin barrier, resulting in inferior protection against bacteria and allergens [5].

It is currently recognized that this disease is found, especially in children and adolescents, within the context of atopy, an entity that also includes asthma, rhino-conjunctivitis, and a significant predisposition to develop allergies [6,7]. Several studies have previously analyzed the association of AD with the presence of other diseases. A systematic review and meta-analysis by Chester et al. in 2021 [8] concluded that AD patients present an increased risk of mental and autoimmune diseases. The narrative review by Paller et al. in 2018 [9] showed that the global burden of AD is associated with mental illnesses such as depression, anxiety, and suicidal ideation, as a result of lack of sleep, itching, and stigmatization due to their skin lesions, both in children and adults. The cohort study by Mortz et al. in 2015 [10], on the other hand, showed that AD is associated with infections, neuropsychiatric disorders, metabolic syndrome, autoimmune diseases, and cancer, among others. Furthermore, the cross-sectional study by Gilaberte et al. in 2020 [11] showed that 43% of children under 18 years of age with AD in Spain have at least one additional comorbidity. The most frequent comorbidities in this study were asthma, psychosocial disorders, and visual disturbances, whereas asthma, allergic rhinitis, and irritable bowel syndrome showed the greatest strength of association with AD.

A better knowledge of the comorbidities surrounding AD could help us guide the care of these patients from a holistic perspective and better understand the etiopathogenesis of this disease. However, chronic diseases rarely appear in isolation and tend to cluster together in the form of disease patterns, which represent non-random associations among diseases. Their study could allow us to identify profiles of AD patients with specific care needs and specific preventive actions, and could also shed some light on the underlying physio-pathological mechanisms.

In this context, network science is a powerful tool that applies clustering techniques that allow us to exhaustively analyze and visualize the associations between diseases to identify disease patterns [12]. Network analysis has already been applied to study the associations among diseases in patients with specific index conditions with relevant clinical results [13,14]. However, to our knowledge, this research approach has not been applied to the study of AD comorbidity.

This study aims to explore the existence of comorbidity patterns in patients with AD using network analysis and to clinically describe the clusters obtained.

## 2. Materials and Methods

We conducted a retrospective, observational study in the EpiChron Cohort, which links socio-demographic and clinical data from all the users of the public health system of the Spanish region of Aragón [15]. This cohort is based on the information registered in the electronic health records (EHRs) and clinical–administrative databases of approximately 98% of the citizens of the region (reference population: 1.3 million people). For this study, we selected all the 50,801 individuals from the cohort diagnosed with AD at some point from 1 January 2010 to 31 December 2018.

The Clinical Research Ethics Committee of Aragón (CEICA) approved this study (Research protocol PI20/633) and waived the requirement to obtain informed consent from patients given the epidemiological nature of the project and the use of anonymized data.

For all patients, we studied sex, age interval (0–2, 3–10, 11–17, 18–65, and >65 years), and all chronic diseases registered in their EHRs. Diagnoses were initially coded using the International Classification of Primary Care, First Edition (ICPC–1), or the International Classification of Diseases, Ninth Revision, Clinical Modification (ICD–9–CM). Subsequently, using the open-source algorithm Chronic Condition Indicator (CCI) [16], each ICD9 code was classified as either chronic or not. The software defines “chronic” as diseases with a duration equal to or greater than 12 months and meeting at least one of the following criteria: (a) require continuous care, that have a high risk of recurrence, and/or that continue to have implications for the management of the patient; (b) imply limitations on self-care, social interactions, and independent living. Once selected, those chronic diagnoses were grouped in 153 clinical categories through the Clinical Classifications Software (CCS) [17] based on the clinical, therapeutic and diagnostic similarities of the diseases.

First, a descriptive analysis of the demographic characteristics of the study population was performed. We summarized the results as proportions for categorical variables and as means and standard deviations for continuous variables.

Then, we performed a network analysis to study the associations between comorbidities of AD. We stratified the population by sex and age interval and built a network for each stratum, with ten networks in total. To increase the clinical interest of the study and to facilitate the interpretation of the results, only diseases with a prevalence > 1% were included in the analysis.

In a disease network, a node represents a disease, and an edge means a statistically significant correlation between a specific pair of conditions. We calculated the tetrachoric correlations of each pair of comorbidities to analyze the weight of the association between them [18]. We used a cut-off point for statistical significance of *p*-value < 0.01 to correct the family-wise error rate due to multiple comparisons [13,14].

Then, we used the network’s modularity to search for clusters of diseases within each network based on the Louvain method [19], as previous disease pattern studies have done [13,14,20]. Modularity analyzes the number of edges in the network, comparing the density of edges inside a group to edges between groups [19]. The Louvain method optimizes the modularity in an iterated process, detecting communities or clusters of diseases. Community detection methods, such as the Louvain or Leiden algorithms, among others, allow the network’s structure to determine the number and size of the cluster obtained [21,22] based on the density of edges and their weight (measured by the tetrachoric correlation) and not by the researcher.

Once we obtained the patterns of diseases for each age and sex stratum, all clinicians named the patterns by consensus. This last step was performed considering the prevalence and clinical relevance of the diseases, and the weight of the tetrachoric correlations, in line with the names already given in the literature.

We performed all the analysis in RStudio software (version 1.4.1106, Rstudio, Boston, MA, USA) and GEPHI software (version 0.9.2).

## 3. Results

### 3.1. Characteristics of the Population

We analyzed a population of 50,801 patients with AD (46.3% men). The demographic characteristics are shown in Table 1. The overall prevalence of AD in Aragon was 3.83%.

The most prevalent diseases found in patients with AD were respiratory (i.e., upper respiratory infections, asthma, and rhinitis), cardio-metabolic (i.e., hypertension, dyslipidemia and obesity), cardiovascular (i.e., cardiac dysrhythmia and coagulation disorders), and mental health diseases (i.e., anxiety and mood disorders). Diseases were combined into seventeen patterns with some sex and age specificities, which are summarized below. The complete output of the analysis is available as Appendix A in which we detailed the complete pattern analysis.

### 3.2. Comorbidity Patterns in Men

We identified fifteen patterns in men, classified as upper respiratory infections, respiratory, otorhinolaryngological (ORL), upper respiratory infections-ORL, congenital anomalies-mental, respiratory-allergic, sensitive-digestive, sensitive-metabolic, headache-mental, cardiometabolic, mental, cardiovascular, dyslipidemic, and geriatric. Their composition, disease prevalence, and correlation between conditions are described below and in Figure 1.

In children aged 0–2, we found three patterns. The *upper respiratory infections* pattern included diseases such as upper respiratory infections, which was the most prevalent condition in this network, esophageal disorders, and congenital anomalies. We found a *respiratory* pattern that had asthma and other upper respiratory diseases, among others. An *ORL* pattern was also described with otitis as its most prevalent disease.

Four patterns in boys aged 3–10 years were identified. One that combined most of the diseases from upper respiratory infections and ORL diseases in children aged 0–2. We found a *respiratory-allergic* pattern that included asthma, other upper respiratory diseases, and rhinitis. The correlation between these two last conditions was almost perfect, with a strength of the correlation of 0.99 out of 1. The *congenital anomalies-mental* pattern associated congenital anomalies, miscellaneous mental health disorders, and other skin inflammatory conditions. The last pattern found was a *sensitive-digestive* pattern which included blindness as the most prevalent condition.

Four other patterns were identified in boys aged 11–17. The *upper respiratory infections-ORL* and *respiratory-allergic* were similar to the previous in children aged 3–10. The *headache-mental* pattern included headache as the most prevalent condition; it also included anxiety and miscellaneous mental health disorders, among other diseases. The *sensitive-metabolic* pattern had diseases such as blindness and vision defects, thyroid disorders, obesity, and other nutritional/endocrine/metabolic diseases.

In men aged 18–65 years, three patterns were identified. One pattern combined most of the diseases from the *upper respiratory-ORL* pattern and the *respiratory-allergic* pattern from boys aged 11–17. A *cardiometabolic* pattern was the cluster with more diseases included, highlighting hypertension, obesity, other nutritional/endocrine/metabolic disorders, dyslipidemia, other inflammatory conditions of the skin, and diabetes. We also identified a *mental* pattern which included screening and history of mental health codes, anxiety, mood disorders, substance-related, and alcohol-related disorders.

In men aged 66 and older, five patterns were detected. The *upper-respiratory-allergic* pattern was similar to that found in men aged 18–65, but included other highly prevalent diseases such as osteoarthritis or cataracts. We found a *cardiometabolic* pattern with hypertension, diabetes, other nutritional/endocrine/metabolic disorders, obesity, COPD, and neoplasms as the most prevalent diseases. We identified a *cardiovascular* pattern that included cardiac dysrhythmias as its most prevalent condition. A *geriatric* pattern, with hyperplasia of the prostate, urinary incontinence, or dementia, among other diseases, was also identified. The last and less specific pattern described included dyslipidemia as its most prevalent disease.

### 3.3. Comorbidity Patterns in Women

We identified eleven patterns in women, which were referred as upper respiratory infections, ORL, upper respiratory infections-ORL, respiratory-allergic, sensitive-digestive, menstrual-dysphoric-metabolic, sensitive, upper respiratory-allergic, cardiometabolic, cardiovascular, and geriatric. Their composition, disease prevalence, and correlation between conditions are described below and in Figure 2.

In girls aged 0–2, the two patterns found were similar to those found in boys aged 0–2 years: *upper respiratory infections* and *ORL*.

We described three patterns in girls aged 3–10, again very similar to the clusters found in boys at the same age: an *upper respiratory infections-ORL* pattern, a *respiratory-allergic* pattern, and a *sensitive-digestive* pattern.

Four other patterns were identified in girls aged 11–17: an *upper respiratory infections-ORL* pattern; a *respiratory-allergic* pattern; a *menstrual dysphoric-metabolic* pattern that included menstrual disorders as the most prevalent condition, but also anxiety, miscellaneous mental health disorders, obesity, thyroid and other metabolic disorders; and finally, the *sensitive* pattern, which included blindness as its most prevalent disease.

In women aged 18–65 years, we found three patterns. The *upper respiratory-allergic* pattern included respiratory diseases such as rhinitis, other upper respiratory diseases, and asthma, but also menstrual disorders and anxiety, among others. A *cardiometabolic* pattern was found, that was mainly characterized by thyroid diseases, obesity, and hypertension but also mood disorders. We also identified *cardiovascular* a pattern that only included three conditions: coagulation and hemorrhagic disorders, cardiac dysrhythmias, and female infertility.

In women aged 66 and older, three patterns were detected. The *upper respiratory-allergic* pattern also included osteoarthritis, osteoporosis, thyroid, and mood disorders, among other diseases. The *cardiometabolic* pattern was found, characterized by hypertension, but also including diabetes mellitus, heart failure, COPD and obstructive sleep apnea. Finally, a *geriatric* pattern that included urinary incontinence, dementia, neoplasms, and chronic skin ulcer as the most prevalent diseases was found.

## 4. Discussion

This study explored the comorbidity patterns of AD through the analysis and visualization of the existing disease networks. Different clusters defined as upper respiratory infections with ORL diseases, respiratory disorders with allergic conditions, and cardiovascular diseases with metabolic disorders, among others, were identified depending on age and gender. These epidemiological findings can be helpful to guide AD patients in the primary, secondary, or even tertiary prevention of their comorbidities and understand their physio-pathological mechanisms.

The present investigation shows the infectious respiratory pattern as the most consistently described pattern across all age groups and sexes. Its main component was respiratory infectious diseases, but its weight decreased in older groups in favor of diseases with an allergic component. Asthma or allergic rhinitis are some of the disorders that have been added. In the youngest groups, this allergic component has a distinct pattern by itself. In the case of children aged 0–2 years, respiratory infections are associated with genital, esophageal, or other malformations, although this is not the case in girls.

The higher incidence of infections in patients with AD has been widely described. Dysfunction of the epithelial barrier, colonization of the skin by *Staphylococcus aureus*, and the use of immunosuppressive drugs are some of their causes. In this context, the Swiss BAMSE cohort revealed a higher incidence of pneumonia, otitis media and antibiotics use in AD patients aged 0–2 years [23]. This fact is consistent with the patterns described. Although patients with AD are colonized by *S. aureus* in up to 70% of the cases and are more likely than the general population to suffer impetigo, herpetic eczema or molluscum contagiosum [24], in our analysis skin infections did not play a relevant role or were associated with extracutaneous infections.

Cardiometabolic diseases have been associated with AD in various epidemiological studies, although this association is less clear than in other diseases such as psoriasis [8,25,26]. Multifactorial etiology has been used to explain this association: insomnia, obesity, diabetes and smoking, among other variables [27,28]. Our study found a pattern of cardiometabolic comorbidities that included hypertension, obesity, and mood and thyroid disorders, among others. This pattern was common in men and women over 18 years of age, although there were differences, including mood and endocrine disorders occurring more frequently in women. As for chronic obstructive pulmonary disease, it was more frequent in the group older than 65 years old for both sexes. The existence of this pattern confirms that patients with AD present comorbidities that are cardiovascular risk factors and that tend to be associated throughout life.

Anxiety, insomnia, and mood disorders, among other mental health problems, are comorbidities with a higher incidence in patients with AD. Recent studies have shown that the earlier the disease appears, the greater the risk of suffering from psychiatric comorbidities is [11,24,29]. We found a pattern of mental comorbidities that appeared in boys older than two years and is maintained up to 65 years of age. In the case of girls, this pattern appears intermingled with a menstrual disease first and metabolic comorbidities that we call menstrual-dysphoric-metabolic pattern. The different ways of interacting between the sexes with the environment and the interpersonal relationships they establish at an early age could explain this phenomenon [30].

In patients over 65 years of age, a geriatric pattern was found in both sexes. This pattern grouped diseases such as urinary incontinence, Parkinson’s disease, dementia, skin ulcers and neoplasms, among others. The association of these diseases, typical of physiological aging, with AD is complex. An increase in neoplasms has been described in patients with AD, with lymphomas being the most strongly associated [9]. Regarding the rest of diseases of this pattern, to our knowledge no studies support a higher incidence of these diseases in patients with AD.

Regarding the limitations of this study, the fact that the clinical information obtained in the EHRs was not originally designed for research could create over- and under-diagnosis of some chronic disorders. Another limitation is the cross-sectional retrospective nature of the study, which does not allow us to know the longitudinal characteristics of the population. Additionally, we have to consider the lack of some variables that could help us explain the results obtained, such as lifestyle information, socioeconomic factors, information on functional status, and analytical variables, among others.

One of the principal strengths of our research is that it was conducted on a population-based cohort, including 98% of the reference population. Moreover, data in the EpiChron Cohort undergo continuous quality control checkups that ensure their accuracy and reliability for research purposes. Another important strength is the innovative method applied to understand comorbidities in AD. Network analysis studies the interrelations between diseases and how patterns emerge from them. This paper shows the potentiality of applying this method to study and visualize the comorbidities of AD and achieve a more holistic understanding of these patients. In this sense, it is also important to highlight that this study exhaustively analyzed all chronic diseases obtained from the patient’s EHRs created by health professionals, and not just the most relevant, prevalent or self-reported diseases.

## 5. Conclusions

We identified similar disease patterns in men and women with AD, with the number and complexity of such patterns increasing with age. This is the first study to analyze the comorbidity patterns of AD patients, and our results can help to guide caregivers of AD patients in the prevention of their comorbidities and to understand the physio-pathological mechanisms underlying the comorbidity patterns identified. This study opens an innovative approach to analyze and help AD patients, although further studies are needed to validate the results obtained in different clinical settings and populations.

## Figures and Tables

**Figure 1 jcm-11-06413-f001:**
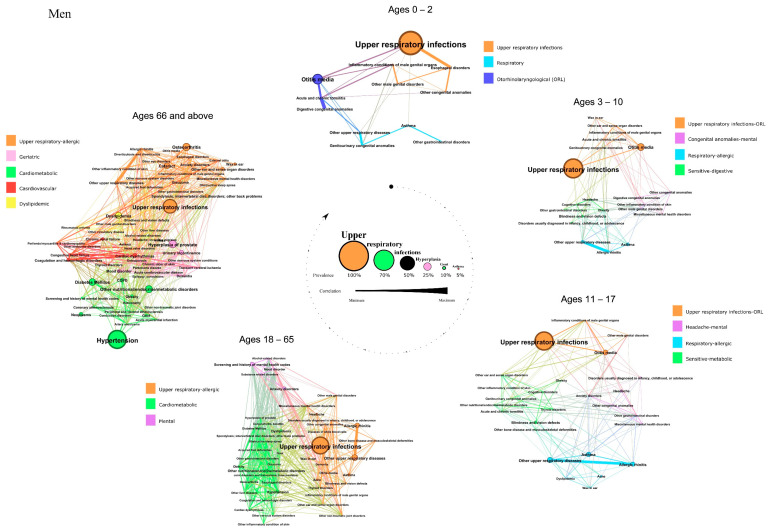
Comorbidity patterns in the networks of men with AD based on age. The diameter of each node and the label size are proportional to the disease prevalence. The width of each link is proportional to the correlation between disease. The colors of the nodes correspond to different patterns.

**Figure 2 jcm-11-06413-f002:**
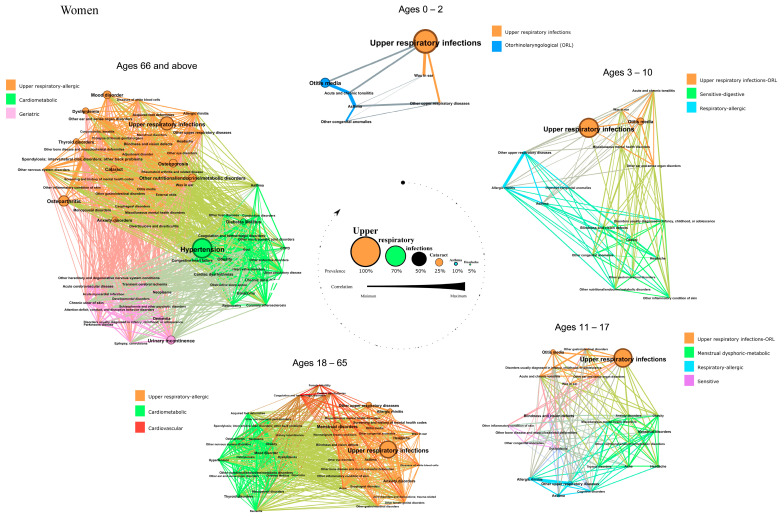
Comorbidity patterns in the networks of women with AD based on age. The diameter of each node and the label size are proportional to the disease prevalence. The width of each link is proportional to the correlation between disease. The colors of the nodes correspond to different patterns.

**Table 1 jcm-11-06413-t001:** Demographic characteristics of patients with AD.

Characteristics	Total (N = 50,801)	Men (N = 23,522)	Women (N = 27,279)
Age interval (N, %)			
0–2 years	3005 (5.9%)	1605 (6.8%)	1400 (5.1%)
3–10 years	17,797 (35.0%)	8954 (38.1%)	8843 (32.4%)
11–17 years	10,955 (21.6%)	54,318 (22.6%)	5637 (20.7%)
18–65 years	14,707 (29.0%)	5985 (25.4%)	8722 (32.0%)
>65 years	4337 (8.5%)	1660 (7.1%)	2677 (9.8%)
Area of Residence (N, %)			
Urban	31,650 (62.3%)	14,627 (62.2%)	17,023 (62.4%)
Rural	19,151 (37.7%)	8895 (37.8%)	10,256 (37.6%)

## Data Availability

The data used in this study cannot be publicly shared, because of restrictions imposed by the Aragon Health Sciences Institute (IACS) and asserted by the Clinical Research Ethics Committee of Aragon (CEICA, ceica@aragon.es). The authors can establish future collaborations with other groups based on the same data. Potential collaborations should be addressed to the Principal Investigator of the EpiChron Group, Alexandra Prados-Torres, sprados.iacs@aragon.es.

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
