# Peer review of "Comorbidity Patterns in Patients with Atopic Dermatitis Using Network Analysis in the EpiChron Study"

_jcm, 2022, doi:10.3390/jcm11216413_

Round 1

Reviewer 1 Report

Thank you for the opportunity to review the article. The article describes an important topic of atopic dermatitis prevalence and related comorbidities. This is especially important as the prevalence of AD is becoming more and more frequent.

Authors should adjust the article formatting to the Journal's requirements. From the abstract to the main text, in-text citations and references need improvement.

The introduction should be enriched with the etiology of AD.

Figures 1 and 2 require enlargement – they are completely unreadable.

The authors should add information on the basis of the diagnosis of AD.

Line 247 – should be Staphylococcus aureus instead of S. aureus.

There are few references to other studies in the discussion – it needs to be expanded. The authors present many results that, in my opinion, are poorly discussed.

Once introduced, abbreviations should be used consistently throughout the article.

Reviewer 2 Report

The authors present a perfectly written study which contains a sophisticated statistical method and network analysis. The manuscript is carefully prepared, both introduction, results part and discussion are of high quality. The study helps to gain a better understanding of comorbidities of AD on a population-wide scale.

Author Response

Dear Reviewer,

On behalf of my collegues, I would like to appreciate your comments. We hope that this manuscript can be published soon to contribute to the knowledge of atopic dermatitis.

Thank you for any time and effort you took in reviewing our manuscript.

Almenara Blasco M, M.D      

Reviewer 3 Report

This research manuscript analyzes comorbidity pattern in AD patients from Aragon, Italy, using network analysis. This method is very new in this field. However, it seems to be important to me to compare the AD population to the general population in Aragon.

Comments:

·        For network analysis, please use the terminology node and edge instead of node and link.

·        How did you choose the significance p-vlaue of 0.01 for multiple comparison?

·        Please clarify what you mean with this sentence: Once we obtained the patterns of diseases for each age and sex stratum, all clinicians named the patterns by consensus.

·        Please explain in detail what you show in Figures 1 and 2, including how many percent of AD patients show these ‘comorbidity patterns’, …

·        Also include proper descriptions in the figures (e.g. age, …)

·        Please explain why clusters with the same or similar name contain different conditions.

·        You should also provide information about the prevalence of the ‘comorbidity patterns’ in the general population / non-AD population.

Round 2

Reviewer 1 Report

The authors responded to my comments. The article should be technically improved (typos, missing spaces, still incorrectly formatted references), but it is interesting and can be published.